# ATG 4B Serves a Crucial Role in RCE-4-Induced Inhibition of the Bcl-2–Beclin 1 Complex in Cervical Cancer Ca Ski Cells

**DOI:** 10.3390/ijms222212302

**Published:** 2021-11-14

**Authors:** Fang-Fang You, Jing Zhang, Fan Cheng, Kun Zou, Xue-Qing Zhang, Jian-Feng Chen

**Affiliations:** Hubei Key Laboratory of Natural Products Research and Development, College of Biological and Pharmaceutical Sciences, China Three Gorges University, Yichang 443002, China; fangfangyou2020@163.com (F.-F.Y.); zj031020@163.com (J.Z.); chengf@ctgu.edu.cn (F.C.); kzou@ctgu.edu.cn (K.Z.); happy.xueqing@163.com (X.-Q.Z.)

**Keywords:** RCE-4, PCD, ATG 4B, the Bcl-2–Beclin 1 complex

## Abstract

RCE-4, a steroidal saponin isolated from *Reineckia carnea*, has been studied previously and has exhibited promising anti-cervical cancer properties by inducing programmed cell death (PCD) of Ca Ski cells. Considering the cancer cells developed various pathways to evade chemotherapy-induced PCD, there is, therefore, an urgent need to further explore the potential mechanisms underlying its actions. The present study focused on targeting the Bcl-2–Beclin 1 complex, which is known as the key regulator of PCD, to deeply elucidate the molecular mechanism of RCE-4 against cervical cancer. The effects of RCE-4 on the Bcl-2–Beclin 1 complex were investigated by using the co-immunoprecipitation assay. In addition, autophagy-related genes (ATG) were also analyzed due to their special roles in PCD. The results demonstrated that RCE-4 inhibited the formation of the Bcl-2–Beclin 1 complex in Ca Ski cells via various pathways, and ATG 4B proteins involved in this process served as a key co-factor. Furthermore, based on the above, the sensitivity of RCE-4 to Ca Ski cells was significantly enhanced by inhibiting the expression of the ATG 4B by applying the ATG 4B siRNA plasmid.

## 1. Introduction

Cervical cancer is the third most common gynecological malignant tumor [1]. Clinically, chemotherapy remains the primary therapeutic regime, but the drug resistance and serious side effects often lead to poor treatment expectations. Natural medicines have attracted extensive attention due to their multiple targets and low toxicity. In our previous studies, RCE-4 (Figure 1), a natural candidate drug for cervical cancer isolated from *Reineckia carnea* [2,3,4,5], could induce PCD characterized by apoptosis and autophagy for cervical cancer Ca Ski cells selectively with an IC50 of 4.71 μmol/L. In addition, the tumor inhibition rate for a human cervical cancer xenograft in nude mice attained 69.1% with the extremely low toxicity to normal tissues [6]. These discoveries highlighted the tremendous value of RCE-4 for treating cervical cancer. Now, RCE-4 has been included as the typical ingredient of *Reineckia carnea* in the local Chinese medicinal material standards of Hubei Province [7]. However, some shortcomings of RCE-4 have also been exposed, such as the relatively high dosage and the molecular mechanism underlying its anti-cervical cancer actions not being fully clarified.

Most current anticancer chemotherapy drugs primarily act by activating programmed cell death (PCD) pathways including apoptosis and autophagy in cancer cells [8]. Targeting the process of PCD with different small-molecule compounds has become a promising therapeutic strategy over the last few decades [9,10]. However, tumor cells have developed novel mechanisms for evading chemotherapy-induced PCD, this could be associated with both the autophagy and inhibition of the more common apoptosis cell death pathways [11]. One of the hallmarks of human cancers is the intrinsic or acquired resistance to apoptosis. Evasion of apoptosis can be part of a cellular stress response to ensure the cell’s survival upon exposure to stressful stimuli. Autophagy is also a protective mechanism of tumor cells that can promote the growth of established tumors. Autophagy-related stress tolerance can enable cell survival by maintaining energy production that can lead to tumor growth and therapeutic resistance. Hence, it was necessary to further explore PCD signaling pathways that are drug-induced, which are not only able to give us new insights into the pathogenesis of tumors but also to aid the development of new targeted therapeutic strategies.

With regards to apoptosis and autophagy as the main two forms of PCD, although there are obvious distinctions in intracellular processes, studies have proved the complex interplay between them [12,13,14]. Bcl-2–Beclin 1 complex, a macro-molecular protein consisting of Bcl-2 and Beclin 1 via the BH3 domain, was confirmed to be the core regulator of crosstalk between apoptosis and autophagy and the “master switch” of PCD [15,16]. The dynamic changes in intracellular Bcl-2–Beclin 1 complex molecules regulate the whole process of PCD, including the initiation, progress and termination [17,18]. The process of PCD can be changed when the binding of Bcl-2 and Beclin 1 is blocked or the dissociation of the complex is promoted, accompanied by the content of the intracellular Bcl-2–Beclin 1 complex decreasing. Numerous studies have shown that the Bcl-2–Beclin 1 complex is stable under nutrient-rich conditions, but when there is a lack of nutrients or other external existing stimuli, some signal molecules participate in the regulation of the complex [19,20]. For example, JNK1 could mediate the rapid phosphorylation of Bcl-2 [21,22], Mst1 could regulate the phosphorylation of Beclin 1 and Dapper 1 could promote the formation of the other competitive Beclin 1-related complexes [23,24], and these could lead to the inhibition of the formation of the Bcl-2–Beclin 1 complex. Ultimately, the apoptosis and autophagic flux processes would all be changed accordingly.

All in all, regulation of the Bcl-2–Beclin 1 complex is of great significance for drug development and the treatment of certain diseases, such as tumors, cerebral ischemia and neurodegenerative disease, etc. [25,26]. However, relevant reports focused on targeting the Bcl-2–Beclin 1 complex for developing anticancer drugs or studying the molecular mechanism of anticancer drugs have not been seen yet. Therefore, the present study focuses on the Bcl-2–Beclin 1 complex and takes RCE-4 as a model drug to deeply clarify the molecular mechanisms underlying its anti-cervical cancer effects.

In addition, it was expected to find effective measures to improve the sensitivity on the basis of understanding the molecular mechanisms, and this might provide theoretical support for the future clinical application of RCE-4.

## 2. Results

### 2.1. RCE-4 Induces Time- and Concentration-Dependent Apoptosis and Autophagy in Ca Ski Cells

RCE-4 was found to induce PCD of cervical cancer Ca Ski cells in our previous studies. In the present study, we further confirmed that RCE-4 could induce time- and concentration-dependent apoptosis and autophagy in Ca Ski cells. As shown in Figure 2A,B, Ca Ski cells were treated with RCE-4 for 6, 12 and 24 h. Compared with the control group, the expression levels of the apoptosis-related proteins cleaved caspase-3/-7/-9 and Bax were increased, caspase-3/-9 and Bcl-2 were reduced (*p* < 0.05 and *p* < 0.01). Additionally, the protein expression levels of the autophagy-related proteins, LC3 II and Beclin1 were increased, and the expression of P62 was decreased. As shown in Figure 2C,E, cells treated with RCE-4 were stained with AO/EB, and an obviously nuclear contraction and rupture were observed in a trapezoidal pattern (*p* < 0.01). In addition, the results of flow cytometry also revealed that RCE-4 induced, significantly, apoptosis in Ca Ski cells (Figure 2D,F, *p* < 0.01).

### 2.2. RCE-4 Inhibits the Formation of Bcl-2–Beclin 1 Complex in Ca Ski Cells

Co-IP assay was used to analyze the relative content of the Bcl-2–Beclin 1 complex in Ca Ski cells. Bcl-2 was used as a bait and anti-Bcl-2 monoclonal antibody as IP, Beclin 1 was analyzed as the target protein to evaluate the relative content of the Bcl-2–Beclin 1 complex. In this experiment, Input was used as the positive control and homologous IgG was used as the negative control to exclude nonspecific binding. Thus, the content of Beclin 1 protein reflected the relative content of the Bcl-2–Beclin 1 complex compared with the blank group. When Beclin 1 was used as a bait, the reverse held true as well, the experimental design was similar. Ca Ski cells were treated with RCE-4 of 8 μmol/L for 6, 12 and 24 h, When using anti-Bcl-2 monoclonal antibody as IP, the expression level of Beclin 1 was significantly reduced, indicating that RCE-4 significantly inhibits the formation of the Bcl-2–Beclin 1 complex in Ca Ski cells. The experimental results when using anti-Beclin 1 monoclonal antibody as IP also confirmed the effect of RCE-4 in inhibiting the formation of the complex (Figure 3).

### 2.3. RCE-4 Inhibits the Formation of Bcl-2–Beclin 1 Complex via Various Pathways

Supportive evidence has gradually revealed that various signal pathways co-regulate the formation of the intracellular Bcl-2–Beclin 1 complex. As shown in Figure 4, under external stimulation, intracellular signal molecule JNK1 was activated and phosphorylated, leading to the phosphorylation of Bcl-2 between the BH4 and BH3 domains on multiple residues, including Thr69, Ser70 and Ser87, which resulted in the dissociation of the Bcl-2–Beclin 1 complex [21,22,27]. In addition, Mst1, a pro-apoptotic protein kinase, could promote the formation of the Bcl-2–Beclin 1 complex by phosphorylating Beclin 1 [23]. On the other side, some core complexes involved in the autophagy regulatory network, such as Beclin 1–HMGB-1 complex [28,29,30], Beclin 1– ATG14–Vps34–Vps15 complex [24,31], Beclin 1–Vps34–Vps15–UVRAG complex and Beclin 1–Vps34–Vps15–UVRAG–Rubicon complex [32,33], would competitively bind to Beclin 1, thus resulting in the reduction of free Beclin 1 molecules, thereby preventing the formation of the Bcl-2–Beclin 1 complex.

The results of western blot (Figure 5A,B) showed that the expression of Dapper1, p-Beclin 1, p-Bcl-2 and p-JNK1 significantly increased and Mst 1 obviously decreased after Ca Ski cells treated with RCE-4 of 8 μmol/L for 6, 12 and 24 h, this inhibited the formation of the Bcl-2–Beclin 1 complex (*p* < 0.001). Additionally, the relative contents of Beclin 1–HMGB-1 complex, Beclin 1–ATG 14–Vps34–Vps15 complex, Beclin 1–Vps34–Vps15–UVRAG complex and Beclin 1–Vps34–Vps15–UVRAG–Rubicon complex were all observed to be increased by the Co-IP assay, thus resulting in the inhibition of the combination of Bcl-2 and Beclin 1 molecules (Figure 5C–E; *p* < 0.001).

### 2.4. ATG 4B Plays a Critical Role in the Inhibition of Bcl-2–Beclin 1 Complex Induced by RCE-4

The Co-IP assay was used to analyze the effect of ATG family proteins on the Bcl-2–Beclin 1 complex in Ca Ski cells treated with RCE-4. Input reflected the regulation of RCE-4 on ATG family proteins. IP Bcl-2 reflected that ATG family proteins bound to the Bcl-2–Beclin 1 complex owing to no reports being seen that ATG molecules could bind to Bcl-2 directly. However, IP Beclin 1 was uncertain because some special ATG proteins can bind to Beclin 1 and form other complexes. For example, ATG14 molecules could bind to Beclin 1 and form the Beclin 1–ATG14–Vps34–Vps15 complex. Thus, for IP Beclin 1, it should be analyzed based on the actual conditions.

As shown in Figure 6A–D, (1) for untreated Ca Ski cells (RCE-4 of 0 μmol/L), when both IP Bcl-2 and IP Beclin 1 could capture the specific protein, this indicated that this protein was involved in the formation of the Bcl-2–Beclin 1 complex. Our results showed that the formation of Bcl-2–Beclin 1 complex might require ATG 3/4 B/5/12/14/16 L1 as co-factors, while ATG 7/13 were irrelevant to the formation of the Bcl-2–Beclin 1 complex (*p* < 0.001). (2) For ATG 3/5/12/16L1, the trend of the Input results was that IP Bcl-2 and IP Beclin 1 were all reduced, which indicated that although ATG 3/5/12/16L1 participated in the formation of the Bcl-2–Beclin 1 complex, it had less effect on the formation of the complex in Ca Ski cells treated with RCE-4 (*p* < 0.001). (3) For ATG14, the trend of the Input results was increased, IP Bcl-2 was reduced but IP Beclin 1 was increased. These results indicated that RCE-4 promoted the expression of ATG 14; however, the increased ATG 14 molecules did not participate in the formation of the Bcl-2–Beclin 1 complex, whereas they were competitively bound to Beclin 1 and formed the Beclin 1–ATG 14–Vps34–Vps15 complex (*p* < 0.001). (4) For ATG 4B, the trend of the Input results was increased, contrary to ATG 14, it was interesting to see that IP Bcl-2 was increased but IP Beclin 1 was reduced. First of all, when Ca Ski cells were stimulated by RCE-4, the expression of ATG 4B significantly increased. Secondly, because no reports had been seen that ATG 4B molecules could bind to Bcl-2 or Beclin 1 directly, so the increased ATG 4B molecules were most likely to bind with the Bcl-2–Beclin 1 complex. In addition, considering that the formation of the Bcl-2–Beclin 1 complex was inhibited in Ca Ski cells treated with RCE-4, if ATG 4B molecules could bind to the Bcl-2–Beclin 1 complex directly, we could conclude that the single Bcl-2–Beclin 1 complex bound more ATG 4B molecules compared with the untreated cells, and its purpose might be to maintain the stability of the complex. In short, these results indicated that, whatever the possibility, ATG 4B not only participated in the formation of the Bcl-2–Beclin 1 complex but also played a key role in protecting the complex from dissociation.

### 2.5. RCE-4 Combined with ATG siRNA Enhances the Sensitivity to Ca Ski Cells

Based on the discovery of the important role of ATG 4B for the Bcl-2–Beclin 1 complex, we tried to improve the sensitivity of RCE-4 to Ca Ski cells by regulating the expression of ATG 4B using ATG 4B siRNA. As shown in Figure 7A, RCE-4 combined with ATG 4B siRNA significantly enhanced the proliferation inhibition and greatly improved the sensitivity of RCE-4 to Ca Ski cells, with IC50 from 4.67 μmol/L reduced to 1.37 μmol/L (*p* < 0.001). In addition, in line with our expectations, the results of Co-IP (Figure 7B,C) declared that the knock out of ATG 4B enhanced the inhibition of RCE-4 on the formation of the Bcl-2–Beclin 1 complex (*p* < 0.001). Furthermore, apoptosis and the depolarization of MMP in Ca Ski cells were detected. As illustrated in Figure 7D–G, the knock out of ATG 4B significantly enhanced the RCE-4-induced apoptosis and depolarization of MMP, compared with only RCE-4-treated cells (*p* < 0.001).

## 3. Discussion

Autophagy and apoptosis play significant physiological roles in cellular survival, stress adaptation and the development of tumors [10,25]. They can be regulated by a variety of regulatory elements (such as sphingolipids, MAPk, etc.) and signaling pathways (such as PI3K/Akt), while the Bcl-2–Beclin 1 complex plays a “toggle switch” role in the occurrence and development of apoptosis and autophagy, which determines whether the cell enters the apoptosis or initiates the autophagy process [15].When the formation of the Bcl-2–Beclin 1 complex is inhibited or disrupted, Bcl-2 and Beclin 1 molecules are released and enter mitochondria and endoplasmic reticulum, respectively, and regulate the process of apoptosis and autophagy. In recent years, more and more studies have focused on this complex. Álvaro F. Fernández et al. found that disruption of the Bcl-2–Beclin 1 complex is an effective mechanism to increase autophagy, prevent premature ageing, improve health span and promote longevity in mammals [34]. Some studies have also found that drug-induced tumor cell death is related to the Bcl-2–Beclin 1 complex pathway [18,26,35,36].

In the present study, we demonstrated the effect of RCE-4 on the Bcl-2–Beclin 1 complex using Co-IP assay. The results showed that the relative content of the Bcl-2–Beclin1 complex in Ca Ski cells treated with RCE-4 was significantly reduced, compared with the control group, which indicated that RCE-4 could inhibit the formation of the Bcl-2–Beclin1 complex. The downside was that whether RCE-4 could disrupt the Bcl-2–Beclin 1 complex directly had not been identified due to the limitation of the test conditions.

Furthermore, we explored the mechanism by which RCE-4 inhibited the formation of the Bcl-2–Beclin 1 complex. Multiple pathways were involved in the regulation, including: (1) RCE-4 induced phosphorylation of JNK1, which in turn led to phosphorylation of multiple residues of Bcl-2 located between the BH4 and BH3 domains, thereby blocking the binding to Beclin 1 via the BH3 domain. (2) RCE-4 inhibited the expression of pro-apoptotic protein kinase Mst1, thus the phosphorylation of Beclin 1 was promoted, in the same way, the binding of the two was blocked. (3) Some important molecules (such as HMGB-1, UVRAG, ATG14, Vps15/34, etc.) involved in the autophagy process competitively bound to Beclin 1 to form other complexes, such as Beclin 1–HMGB–1 complex, Beclin 1–ATG 14–Vps34–Vps15 complex, Beclin 1–Vps34–Vps15–UVRAG complex and Beclin 1–Vps34–Vps15–UVRAG–Rubicon complex, which also affected the formation of the Bcl-2–Beclin 1complex. These findings indicated the complexity of the anti-cervical cancer molecular mechanism of RCE-4.

Multiple protein molecules are involved in the crosstalk between apoptosis and autophagy as co-regulatory factors, especially ATG family members [37,38,39] such as ATG 4, which plays a central role in the LC3 lipid conjugation system, essential for the late step of autophagosome formation [40,41,42]. ATG 5 participates in the external activation of apoptosis and can cause cell death with the participation of FADD-mediated caspase enzymes [43,44,45]. ATG 3/7 can be cleaved by the caspase enzyme and lose its ability to induce autophagy, but it can also affect apoptosis by translocating to mitochondria, making cells more sensitive to apoptosis [46,47]. This progress suggests that the ATG family members play important roles in the process of PCD characterized by apoptosis and autophagy, but no previous studies have found whether ATG family proteins are involved in the formation of the Bcl-2–Beclin 1 complex. In this regard, we tried to make a preliminary exploration of this. Our results indicated that ATG family proteins, such as ATG 3/4B/5/12/14/16L1, participated in the formation of the Bcl-2–Beclin 1 complex as co-factors, while ATG 7/13 were not involved; among them ATG 4B played a key role in maintaining the stability of the complex. A more interesting finding was that when Ca Ski cells were stimulated by RCE-4, the expression of ATG 4B significantly increased, which led to more ATG 4B molecules binding to the Bcl-2–Beclin 1 complex to maintain the stability of the complex, this might be a self-protection mechanism of cells under stress conditions. In previous studies, scholars have found that ATG 4B was complicated related to tumor progression. For example, the expression of ATG 4B is significantly increased in colorectal cancer cells, suggesting that ATG 4B may be important for cancer biology [48]. Debra Akin et al. found that ATG 4B had a positive impact on the tumor growth of Saos-2 cells, and osteosarcoma Saos-2 cells lacking ATG 4B failed to survive under amino acid starvation conditions and had attenuated tumor growth in mice [49]. These findings show the possibility of ATB 4B as a novel target for cancer therapy, but the research into the deep mechanism underlying its actions had not been clarified, our findings could provide a new explanation for this.

Based on the above research, we tried to enhance the sensitivity of RCE-4 to Ca Ski cells via knocking out ATG 4B by siRNA. The results demonstrated that RCE-4 combined with ATG 4B siRNA significantly enhanced the proliferation inhibition of Ca Ski cells by inhibiting RCE-4-induced autophagy, enhancing RCE-4-induced apoptosis and strengthening the inhibition of RCE-4 on the formation of the Bcl-2–Beclin 1 complex in Ca Ski cells. In our previous studies, we found that RCE-4-induced autophagy was protective for Ca Ski cells, which could protect Ca Ski cells from apoptosis, and inhibiting autophagy could enhance RCE-4-induced apoptosis and the sensitivity of RCE-4 to Ca Ski cells; this was consistent with the results of this experiment. Debra Akin et al. also showed similar results, the ATG 4B antagonist named NSC185058 effectively inhibited ATG 4B activity in vitro and in cells, inhibited autophagy and had a negative impact on the growth of osteosarcoma tumors [48]. In another study, the inhibition of ATG 4B by siRNA enhanced lupulone derivatives-induced apoptosis in prostate cancer cells [40]. Moreover, blocking autophagy by inhibiting ATG 4B sensitized several types of resistant carcinoma cells, including MDA-MB-231 and A549 cell lines, to radiation therapy [50].

## 4. Materials and Methods

### 4.1. Cell Culture

Human cervical cancer Ca Ski cells were obtained from The Cell Bank of Type Culture Collection of The Chinese Academy of Sciences and maintained in RPMI-1640 culture medium (Gibco/Thermo Fisher Scientific, Waltham, MA, USA) supplemented with 10% fetal bovine serum (Zhejiang Tianhang Biological Technology Co., Ltd, Hangzhou, Zhejiang, China), 0.2% HEPES and 2% double antibody at 37 °C with 5% CO_2_ in a humidified incubator.

### 4.2. Reagent and Antibodies

The RCE-4 used in the present study was a spiral steranol saponin extracted and purified from *Reineckia carnea* [51]. The RCE-4 stock solution of 50 mmol·L^−1^ was prepared using DMSO and diluted to the desired concentrations in RPMI-1640 medium. Rabbit monoclonal antibodies against Beclin 1 (#3495S), Bcl-2 (#4223S), p-Bcl-2 (#2875T; #2827T), Mst 1 (#14946S), ATG 3 (#3415S), ATG 4B (#13507S), ATG 5 (#12994T), ATG 7 (#8558T), ATG 12 (#4180S), ATG 13 (#13468S), ATG 14 (#96752S), ATG 16L1 (#8089S), HMGB-1 (#MAB1690-SP), UVRAG (#NBP2-24482SS), Vps34 (#3358T), Rubicon (#8465S) and β-actin (#8457S) (Cell Signaling Technology, Boston, USA; R&D Systems, Minneapolis, MN, USA) were used at a dilution of 1:1000. HRP-conjugated secondary antibody (1:5000) was purchased fromSanta Cruz Biotechnology, Inc., Dallas, TX, USA.

### 4.3. Cytotoxicity Assay

An MTT assay was used to evaluate the effects of RCE-4 extract on cell growth, as previously described. Ca Ski cells (1 × 10^5^/mL) were seeded into 96-well plates, incubated at 37 °C for 12 h. The Ca Ski cells were pretreated with ATG 4B siRNA for 6 h, and then treated with RCE-4 (0, 0.5, 1, 2, 4, 8, 16 and 32 μmol/L) for an additional 48 h. Subsequently, 20 μL 5 mg/mL MTT (BS0328; Amersco, Spokane, WA, USA) reagent was added per well. After 4 h, the media was gently removed and 150 μL DMSO (Sigma-Aldrich, St. Louis, MO, USA) was added. The absorbance was measured using a microplate reader at 490 nm, subtracting the baseline reading.

### 4.4. Flow Cytometry Assay

Flow cytometry was used to detect the effects of RCE-4 on apoptosis of Ca Ski cells. The GFP-CERTIED^®^ Apoptosis/Necrosis Detection kit (cat. no. ENZ-51002-100; Enzo Life Sciences, New York, NY, USA) allows for easy differentiation of early apoptosis from late apoptosis or necrosis; thus allowing for analysis of the separate death pathways in detail. A blank control was set; Ca Ski cells were pretreated with or without ATG 4B siRNA and then treated with RCE-4 for 24 h. Cells were collected, centrifuged, separated from the supernatant, washed twice with PBS, then sat to dry. The cell suspension was prepared with a buffer/water ratio of 1:9. The appropriate amounts of apoptotic and necrotic staining reagents were added according to the ratio of suspension–Staining solution = 100:1, then mixed gently. The mixture was stained for 15 min at room temperature in the dark and flow cytometry was used to detect staining.

### 4.5. Acridine Orange/Ethidium Bromide (AO/EB) Double Fluorescent Staining

AO/EB double fluorescence staining was used to detect the effects of RCE-4 on the apoptosis of Ca Ski cells. AO can penetrate through the cell membrane and bind to DNA, fluorescing green upon doing so. EB can only bind with DNA if the cell membrane is damaged, giving an orange-red appearance, and the orange-red brightness is higher than the green color of AO. Apoptotic cells are unevenly stained due to the high concentration of chromatin, whereas the non-apoptotic cells have a normal structure and uniform coloring. Observed under a fluorescent microscope, four cell states can be seen: viable cells, bright green chromatin with organized structure; viable apoptotic cells, bright green chromatin, which is highly condensed or fragmented; non-apoptotic nonviable cells, bright orange chromatin with organized structure; nonviable apoptotic cells, bright orange chromatin, which is highly condensed or fragmented. After treatment, the Ca Ski cells were collected and washed with PBS and AO Stain Buffer (1x) once. An appropriate quantity of AO Stain Buffer (1x) was used to resuspend the cells, mixed according to a cell suspension ratio of AO Stain–cell suspension of 19:1, and then an appropriate quantity of AO/EB stain was added and indicated in the dark for 20 min. A fluorescence microscope was used to observe staining and data was analyzed using Image-Pro-Plus version 7.0.

### 4.6. MMP Assay

The effect of RCE-4 on MMP in Ca Ski cells was assessed using JC-1 staining (cat. no. C2006; Beyotime Institute of Biotechnology, Songjiang, Shanghai, China). JC-1 is an ideal and widely used fluorescent probe for the detection of MMP. In normal mitochondria, JC-1 aggregates in the mitochondrial matrix to form polymers, which emits a strong red fluorescence. In damaged mitochondria, JC-1 only exists as a monomer in the cytoplasm and exhibits green fluorescence due to the decrease in or loss of membrane potential. JC-1 can be used not only for qualitative detection but also for quantitative detection as the change in its color can directly reflect the change in MMP since the degree of mitochondrial depolarization can be measured by the ratio of red to green fluorescence intensity. After treatment, Ca Ski cells were incubated with JC-1 dye for 20 min at room temperature, and then washed twice with dyeing buffer. Cells were treated with carbonyl cyanide m-chlorophenylhydrazone CCCP (0.1 μmol/L), and those treated with mitochondrial inhibitors were used as the positive controls. A fluorescence microscope was used to observe the staining. Data were analyzed using Image-Pro-Plus.

### 4.7. Western Blotting

Ca Ski cells were treated with RCE-4 of 8 μmol/L for 6, 12, and 24 h. Total protein was extracted from the cells using a protein extraction kit. Following protein quantification, protein lysates were added to the sample buffer and boiled for 10 min. Subsequently, samples were loaded on a 10% SDS-gel and resolved using SDS-PAGE (electrophoresis at 80 V and 30 mA for 2.5 h). Subsequently, proteins were transferred at 200 mA to PVDF membranes. After blocking at room temperature for 2 h, membranes were incubated with the primary antibodies at 4 °C overnight in 5% skimmed milk. Membranes were washed five times using tributyltin compound plus 0.05% Tween-20, and incubated with the secondary antibody for 1.5 h at 37 °C. Signals were visualized using ECL (Beyotime Institute of Biotechnology, Songjiang, Shanghai, China), and developed using Kodak film and produced using a Tanon 5200 luminescence imaging system (Tanon Technology, Shanghai, China). Densitometry analysis was performed using ImageJ version 2.1 (National Institutes of Health, Bethesda, MD, USA) using β-actin as the control load.

### 4.8. Plasmid Small Interfering (si)RNA Transfection

The plasmids ATG 4B siRNA (h) (cat. no. sc-72584; Santa Cruz Biotechnology, Inc., Dallas, TX, USA), ATG 4B (h)-PR (sc-72584-PR; Santa Cruz Biotechnology, Inc.), and Control siRNAs including Control (cat. no. sc-37007; Santa Cruz Biotechnology, Inc.) and Control siRNA (FITC Conjugate)-A (cat. no. sc-36869; Santa Cruz Biotechnology, Inc.) were obtained from Youning Life Support Technology. The plasmids were transfected into Ca Ski cells according to specifications.

### 4.9. Co-Immunoprecipitation (Co-IP)

After Ca Ski cells were treated with RCE-4 of 8 μmol/L, lysis buffer (cat. no. abs9116-100 ML; Absin, Shanghai, China) was used to lyse the Ca Ski cells. Protein A/G PLUS-Agarose (cat. no. sc-2003; Santa Cruz Biotechnology, Inc.) was used to purify the lysates, which were subsequently mixed with Bcl-2 (cat. no. 4223S; Cell Signaling Technology, Boston, MA, USA) or Normal Rabbit IgG (cat. no. 2729S; Cell Signaling Technology, Inc.) and then shaken for 2 h at room temperature. The immunoprecipitants were captured using protein A/G agarose beads and analyzed by western blotting.

### 4.10. Statistical Analysis

Data are presented as the mean ± standard deviation. Data were analyzed using GraphPad Prism version 5.0 (GraphPad Software Inc, San Diego, CA, USA). Comparisons between groups were assessed using a Student’s *t*-test or a one-way ANOVA. *p* < 0.05 was considered to indicate a statistically significant difference.

## 5. Conclusions

This study demonstrated that inhibition of ATG 4B significantly reduced RCE-4-induced autophagy, enhanced RCE-4-induced apoptosis and strengthened the inhibition of RCE-4 on the formation of the Bcl-2–Beclin 1 complex in Ca Ski cells, thereby enhancing the sensibility of RCE-4 to Ca Ski cells.

## Figures and Tables

**Figure 1 ijms-22-12302-f001:**
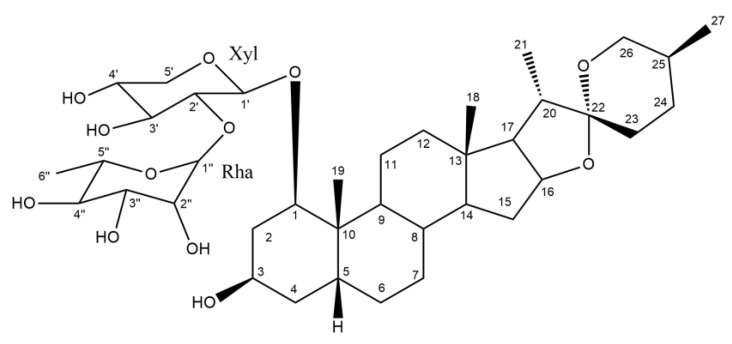
The structure of RCE-4 (1β, 3β, 5β, 25S)-spirostan-1,3-diol1-[α-L-rhamnopyranosyl-(1→2) -β-D-xylopyranoside].

**Figure 2 ijms-22-12302-f002:**
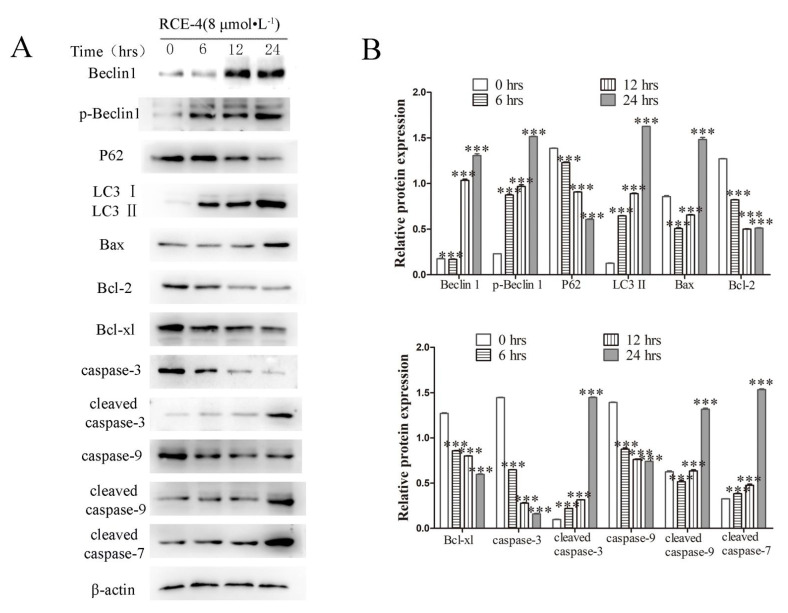
RCE-4 induces time- and concentration-dependent apoptosis and autophagy in Ca Ski cells. (**A**) The protein expression levels of apoptosis associated proteins in Ca Ski cells treated with RCE-4 at different times was analyzed by western blotting. β-actin was used as the loading control. (**B**) Densitometry analysis of the expression of proteins. (**C**) and (**E**) AO/EB Staining was used to detect apoptosis in Ca Ski cells. Scale bar: 100 μM. Viable cells, bright green chromatin with organized structure; viable apoptotic cells, bright green chromatin, which is highly condensed or fragmented; non-apoptotic nonviable cells, bright orange chromatin with organized structure; nonviable apoptotic cells, bright orange chromatin, which is highly condensed or fragmented. Data are presented as the mean ± standard deviation of three independent experiments. (**D**,**F**) Ca Ski cells were treated with RCE-4 for 24 h, and apoptosis was detected by flow cytometry. Data are presented as the mean ± standard deviation of three independent repeats, *** *p* < 0.001 vs. Control. AO, acridine orange.

**Figure 3 ijms-22-12302-f003:**
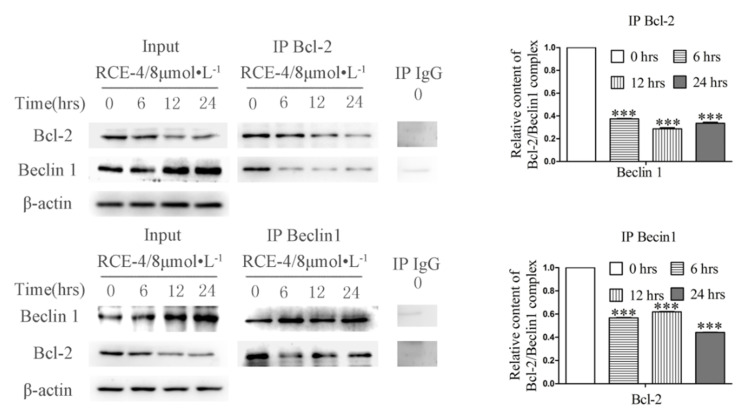
Effect of RCE-4 on the Bcl-2–Beclin 1 complex in Ca Ski cells. Co-IP was used to confirm the effects of RCE-4 on the Bcl-2–Beclin 1 complex in Ca Ski cells. The Ca Ski cells were treated with RCE-4 of 8 μmol/L for 6, 12 and 24 h. Whole cell lysates were purified using Protein A/G PLUS-Sepharose, and then mixed with Bcl-2, Beclin 1 or control IgG antibodies. The immunoprecipitate was captured on Protein A/G PLUS-Agarose and analyzed using western blotting with antibodies against Bcl-2 and Beclin 1. Relative expression of Beclin 1 and Bcl-2 expressed as a percentage of the untreated group, and relative content of the Bcl-2–Beclin 1 complex in Ca Ski cells. Data are presented as the mean ± standard deviation of three independent experiments. *** *p* < 0.001 vs. Control.

**Figure 4 ijms-22-12302-f004:**
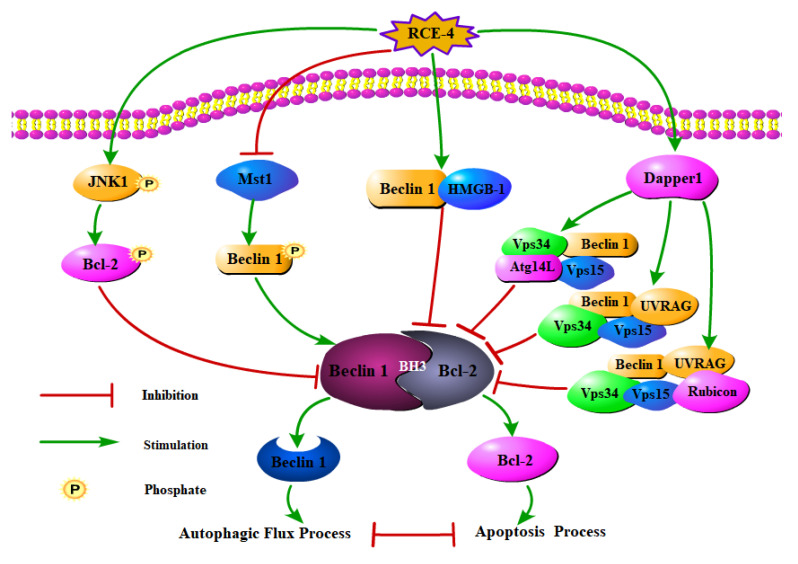
Various signal pathways co-regulated the formation of intracellular Bcl-2–Beclin 1 complex.

**Figure 5 ijms-22-12302-f005:**
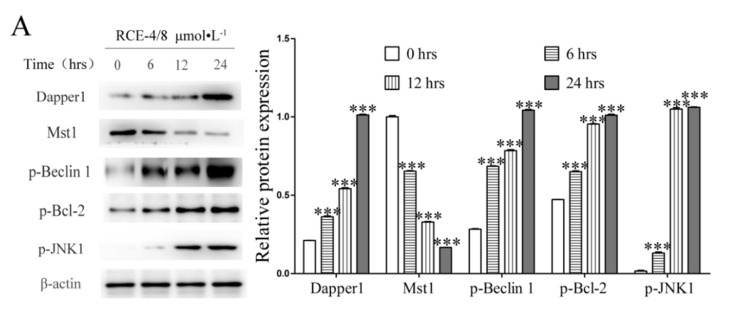
Pathway by which RCE-4 inhibits the formation of the Bcl-2–Beclin 1 complex. (**A**) Protein expression levels in Ca Ski cells treated with RCE-4 at different time points were analyzed by western blotting. β-actin was used as the loading control. (**B**) The content of Beclin 1–HMGB-1 complex was detected by Co-IP assay. Data are presented as the mean ± standard deviation of three independent experiments. *** *p* < 0.001 vs. Control. (**C**) The content of Beclin 1–ATG 14–Vps34–Vps15 complex was detected by Co-IP assay. Data are presented as the mean ± standard deviation of three independent experiments. *** *p* < 0.001 vs. Control. (**D**) The content of Beclin 1–Vps34–Vps15–UVRAG complex was detected by Co-IP assay. Data are presented as the mean ± standard deviation of three independent experiments. ** *p* < 0.01, *** *p* < 0.001 vs. Control. (**E**) The content of Beclin 1–Vps34–Vps15–UVRAG–Rubicon complex was detected by Co-IP assay. Data are presented as the mean ± standard deviation of three independent experiments. *** *p* < 0.001 vs. Control.

**Figure 6 ijms-22-12302-f006:**
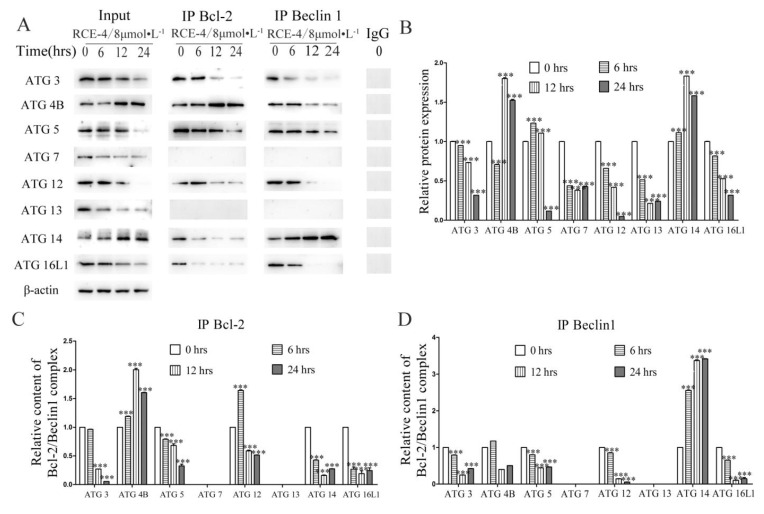
Analysis of the roles of ATG family proteins in the inhibition of Bcl-2–Beclin 1 complex induced by RCE-4. (**A**) The Ca Ski cells were treated with RCE-4 of 8 μmol/L for 6, 12, and 24 h. Following, the Co-IP experiments were performed. (**B**–**D**) Densitometry analysis of the expression of proteins. Data are presented as the mean ± standard deviation of three independent experiments. *** *p* < 0.001 vs. Control.

**Figure 7 ijms-22-12302-f007:**
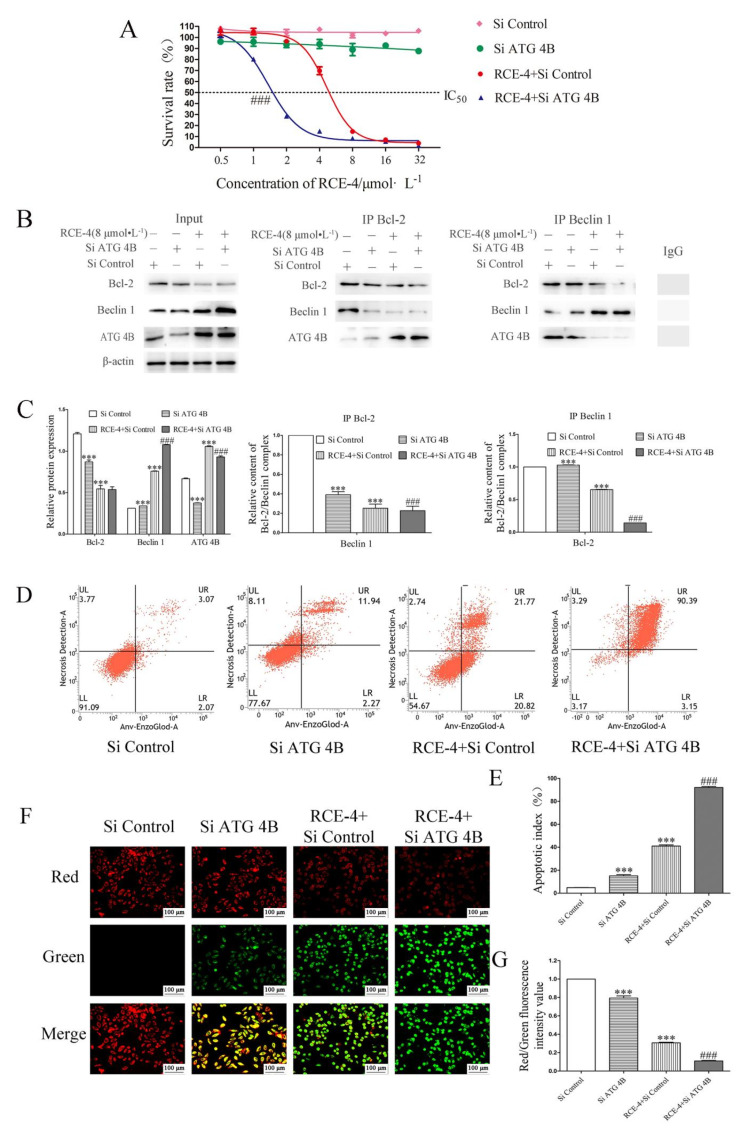
RCE-4 combined with ATG 4B siRNA enhanced the sensitivity of Ca Ski cells. (**A**) RCE-4 combined with or without ATG 4B siRNA inhibited the proliferation of Ca Ski cells. Data are presented as the mean ± standard deviation of three independent experiments. ^###^ *p* < 0.001 vs. RCE-4. (**B**) Co-IP was used to confirm the effects of RCE-4 combined with ATG 4B siRNA on the Bcl-2–Beclin 1 complex in Ca Ski cells. (**C**) Relative expression of Beclin 1 and Bcl-2 expressed as a percentage of the untreated group, and relative content of the Bcl-2–Beclin 1 complex in Ca Ski cells. Data are presented as the mean ± standard deviation of three independent experiments. *** *p* < 0.001 vs. Control. ^###^ *p* < 0.001 vs. RCE-4. (**D**) and (**E**) The effect of RCE-4 combined with ATG 4B siRNA on the apoptosis of Ca Ski cells. Data are presented as the mean ± standard deviation of three independent experiments. *** *p* < 0.001 vs. Control. ^###^ *p* < 0.001 vs. RCE-4. (**F**,**G**) The effect of RCE-4 combined with ATG 4B siRNA on the mitochondrial transmembrane potential of Ca Ski cells. In normal mitochondria, JC-1 aggregates in the mitochondrial matrix to form polymers, which emit a strong red fluorescence. In damaged mitochondria, JC-1 only exists as a monomer in the cytoplasm and exhibits green fluorescence due to the decrease in or loss of membrane potential. Data are presented as the mean ± standard deviation of three independent experiments. *** *p* < 0.001 vs. Control. ^###^ *p* < 0.001 vs. RCE-4.

## Data Availability

The original data of the graphics in the paper can be obtained by contacting the main author.

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
