# Peer review of "ATG 4B Serves a Crucial Role in RCE-4-Induced Inhibition of the Bcl-2–Beclin 1 Complex in Cervical Cancer Ca Ski Cells"

_ijms, 2021, doi:10.3390/ijms222212302_

Round 1

Reviewer 1 Report

The manuscript written by You et al. is definitely interesting and the author performed quite high number of experiments to prove their thesis. Nevertheless, some improvements are required.

Major remarks:

In Figure 2C the images must be self -explanatory and the colors meaning must be included in the picture and legend (the same is true for 7F

In Figure 2D authors should include the proper description of both axis, early apoptotic cells, late apoptotic cells and necrotic ones should be distinguish by the proper interpretation of the graph. Based on the axis and methods description it is not possible to understand what was measured and what is the difference between 4 cells populations presented in the graph. ( the same is true for 7D)

Figure 4 seems not to be complete (right side)

 Based on the data presented in  Figure 6A it is not clear for me why authors exclude the potential formation of ATG4B-BCL-2 complex. The hypothesis  about complex protection by ATG4B seems not to be supported by the data. If ATGB4 binds to the complex how the authors explain the data with IP-Beclin1 and ATG4B (Figure 6A) ?

In Figure 7B the design of experiment is not clear for me, authors compare Si Control and SiControl/SiATG 4B, whereas after treatment with RCE-4 authors compare no si with siATG 4B,  the same is true for IP-Bl-2   and IP-Beclin1.  

In Figure 7A is also not clear if the authors used control siRNA or not. Moreover, since the KD is not convincing based on Western Blot the information about transfection efficiency should be included.

Minor remarks:

Very often in the text the capital letter appear in the middle of the sentence, there are also very common typo mistakes

Author Response

尊敬的审稿人,请查看附件!

Reviewer 2 Report

The authors indicate interesting findings of RCE-4 function. However, there are several problems to make clear.

Major Concerns

The authors indicated phosphor-JNK that, however without indication JNK amount. So, it is unclear JNK and phosphor-JNK increase or just increase of phosphorylation of JNK. They should indicate JNK data.

In Fig.4 and the sentence “Mst1, a pro-apoptotic protein kinase, could also disrupt the formation of the Bcl-2-Beclin 1 complex by phosphorylating Beclin 1” is perfectly contradict the original reference# 30. As described in the title “Mst1 inhibits autophagy by promoting the interaction between Beclin1 and Bcl-2.” in L134-5.  In addition, if RCE-4 can inhibit Mst1 as in FIG4, phosphor-Beclin should decrease. In Fig.4 the mark from phospho-Beclin to Beclin1-Bcl2 complex should be an arrow.  

In Fig. 5after RCE4 treatment Mst1 decrease however phospho-Beclin increase with Beclin amount.  Hopefully, they indicate Beclin and phospho-Beclin together in the same panel.

If the authors believed RCE-4 can inhibit ATG4B function they should didn’t indicate the data overexpressing of ATG4B suppress of RCE-4 function. 

There is no clear explanation about the finding ATG4B inhibition could induce apoptosis.

I hope the authors should indicate some hypotheses. 

The data is limited to Ca Ski cells.  It is unclear why the authors chose this cell line. 

Minot concerns

The concentrations indicated as “μmol●L-1” in the manuscript should be replaced “μmol/L”.

The left and bottom sides of FIG.4 are disrupted in the PDF file. The size of the image should be appropriately adjusted to the page.    

“deeply” should be replaced “deep” in L284 “the research of deeply mechanism underlying its actions had not been clarified”

Author Response

Dear reviewer, please check the attachment!

Round 2

Reviewer 2 Report

I recognised the authors can sincerely make revised version of the manuscript and correct it as much as they can. I agree to publish the manuscript in IJMS.